# Coping Strategies and Perceiving Stress among Athletes during Different Waves of the COVID-19 Pandemic—Data from Poland, Romania, and Slovakia

**DOI:** 10.3390/healthcare10091770

**Published:** 2022-09-14

**Authors:** Ryszard Makarowski, Radu Predoiu, Andrzej Piotrowski, Karol Görner, Alexandra Predoiu, Rafael Oliveira, Raluca Anca Pelin, Alina Daniela Moanță, Ole Boe, Samir Rawat, Gayatri Ahuja

**Affiliations:** 1Faculty of Administration and Social Sciences, Academy of Applied Medical and Social Sciences in Elbląg, 82-300 Elblag, Poland; 2Faculty of Physical Education and Sport, National University of Physical Education and Sports, 060057 Bucharest, Romania; 3Department of Personality Psychology and Forensic Psychology, University of Gdańsk, 80-309 Gdańsk, Poland; 4Faculty of Sports, University of Presov, 08001 Presov, Slovakia; 5Sports Science School of Rio Maior—Polytechnic Institute of Santarém, 2040-413 Rio Maior, Portugal; 6Research Centre in Sport Sciences, Health Sciences and Human Development, 5001-801 Vila Real, Portugal; 7Life Quality Research Centre, 2040-413 Rio Maior, Portugal; 8Department of Physical Education and Sports-Kinetotherapy, Faculty of Medical Engineering, University Politehnica of Bucharest, 060042 Bucharest, Romania; 9Department of Business, Strategy and Political Sciences, University of South-Eastern Norway, 3045 Drammen, Norway; 10Institute of Psychology, Oslo New University College, 0456 Oslo, Norway; 11Military MIND Academy, Pune 411060, India; 12Department of Education, Ali Yavar Jung National Institute of Speech and Hearing Disabilities, Mumbai 400050, India

**Keywords:** sport, athlete, coping with stress, SARS-CoV-2, COVID-19, cross-cultural research

## Abstract

Coronavirus disease (COVID-19), an infectious disease caused by the SARS-CoV-2 virus, has affected numerous aspects of human functioning. Social contacts, work, education, travel, and sports have drastically changed during the lockdown periods. The pandemic restrictions have severely limited professional athletes’ ability to train and participate in competitions. For many who rely on sports as their main source of income, this represents a source of intense stress. To assess the dynamics of perceived stress as well as coping strategies during different waves of the COVID-19 pandemic, we carried out a longitudinal study using the Perception of Stress Questionnaire and the Brief COPE on a sample of 2020 professional athletes in Poland, Romania, and Slovakia. The results revealed that in all three countries, the highest intrapsychic stress levels were reported during the fourth wave (all, *p* < 0.01) and the highest external stress levels were reported before the pandemic (*p* < 0.05). To analyze the data, analyses of variance were carried out using Tukey’s post hoc test and η2 for effect size. Further, emotional tension was the highest among Polish and Slovak athletes in the fourth wave, while the highest among Romanian athletes was in the pre-pandemic period. The coping strategies used by the athletes in the fourth wave were more dysfunctional than during the first wave (independent t test and Cohen’s d were used). The dynamics of the coping strategies—emotion focused and problem focused—were also discussed among Polish, Romanian, and Slovak athletes. Coaches and sports psychologists can modify the athletes’ perceived stress while simultaneously promoting effective coping strategies.

## 1. Introduction

SARS-CoV-2 (severe acute respiratory syndrome coronavirus 2) caused the coronavirus disease 2019 (COVID-19), which was classified as a pandemic by the World Health Organization in March of 2020, only a few months after the first case was detected in December of 2019 [1]. The dynamics of transmission, severe health consequences, and high mortality of infected patients were reported in many societies. Thus, many governments decided to introduce restrictions aimed at limiting the transmission of the virus [2]. Personal protective measures, such as face masks, frequent hand disinfecting, and social distancing, proved insufficient to stop the pandemic entirely. Many countries also introduced lockdowns, which forced people to stay home, thus, limiting physical and sports activity [3].

The lockdowns were intended to protect against infection but, unfortunately, they resulted in negative health consequences, such as weight anxiety and depression increases [4,5]. Individual countries tried to limit the spread of COVID-19 by means of various restrictions and staying indoors (people being confined to their homes), situations that impacted the general population’s mental health (people did not have an adequate space in which to work or exercise) [6,7]. It was assumed that COVID-19′s droplet transmission may be facilitated by strenuous physical activity, which results in deep lung ventilation [8]. Due to the lockdowns, sports halls and gyms were closed and outdoor physical activity was banned. Many people who were physically active were forced to suspend training almost overnight. Studies on athletes from over 140 countries showed that lockdowns led to lower intensity, frequency, and duration of training [9]. For professional athletes, lockdowns led to violations in long-term, rigorous training plans, inability to prepare for competitions, and canceling or severely limiting sports events [10]. For some of them, this may have caused an early end to their career due to their age and the inflexible schedules of global events [11]. The COVID-19 crisis decreased athletes’ functional psychobiosocial states, e.g., cognitive, emotional, motivational, volitional, motor-behavioral, communicative, etc. (for psychobiosocial state examples, see [12]), while increasing their dysfunctional psychobiosocial states [13]. Even during the rebooting of sport activities, despite the resumption of sport activities, athletes experienced a detrimental situation, their mental health being still affected [14]. Further, increased stress and anxiety were common themes affecting student athletes’ experiences when returning to sport amidst the COVID-19 pandemic [15].

Professional athletes who rely on sports as their main source of income found themselves in a difficult situation, as their physical activity was reduced and, thus, their income from broadcasting competitions, advertising, and awards dropped to almost zero. Restricting sports activities had a negative effect on athletes’ health, but it was not effective in preventing the spread of COVID-19 in this population group [16]. Athletes’ mental status naturally influences their functioning. A global sense of threat, social isolation, and uncertainty about the future may lead to anxiety, depression, and chronic stress [17].

Effective coping strategies reduced the stress experienced by athletes during the COVID-19 pandemic [15,18]. More specifically, positive reframing helps athletes to maintain a positive mood state and reduce distress, while self-blaming and behavioral disengagement are coping strategies that negatively influence athletes’ mood. Regarding coping, it represents the conscious use of affective, cognitive, or behavioral efforts to effectively deal with demands, events that the individual perceives as potentially harmful or unpleasant [19]. The outcome of coping efforts is to reduce psychological distress, improve mental well-being and reduce physiological reactions that may impair performance [20,21]. The coping literature [22,23] discusses identifying the athlete’s cognitive appraisal regarding the situation in which he/she is found (in training or competition). Larazus and Folkman [23] categorized appraisal as non-stressful (positive, harmless) or stressful, while stress appraisals were designated as threatening or challenging. Hoar et al. discussed (within another appraisal framework) the importance of perceived control. As authors mentioned, taking into account that perceived control can change over time, coping responses can become more or less effective (even well-learned coping strategies can be modified, as a consequence of environmental demands) [24]. Considering problem-focused and emotional approach coping, specialized literature underlines that men engage in more problem-focused coping, whereas women resort to more emotional-approach coping [25]. As stated, men who are using more emotion-focused coping strategies seemed to register higher levels of positive affect.

During the COVID-19 pandemic, physical activity can “generate and maintain resources combative of stress and protective of health” [26]. Significantly lower mental and physical health was found in individuals with the highest decrease in physical activity during the pandemic. Considering professional cyclists, those who followed a Sport Psychology Intervention (online) during the pandemic coped better with sport psychological stressors (no significant improvements were found, however, for sport social and for sport emotional well-being factors) [27]. In order to cope with negative psychological effects arising from the pandemic, researchers discussed the benefits of mental toughness training in the case of athletes [28], important aspects, since professional athletes obtained lower values for the agreeableness factor (during the pandemic crisis), compared to non-professionals [29]. Counselling athletes in an unprecedented situation, as in the COVID-19 pandemic, is very important for acquiring healthy behaviors. For example, mindful activities related to the body—the experience of one’s body as trustworthy and safe—could reduce distress in athletes and increase positive stress [14]. Training regimens should be introduced as standard habits for well-being and health, especially for women and novice athletes, who registered higher levels of negative stress (distress) [13].

Each country is characterized by different dynamics of COVID-19 infections, resulting from, among others, the implemented prevention methods, the number of social contacts and foreign travels, national age average, healthcare quality, and economic conditions.

The purpose of the current research was to investigate the dynamics of stress perceived by Polish, Romanian, and Slovak athletes during the first four waves of the COVID-19 pandemic and to establish the changes in coping strategies between the first and fourth waves. The following research questions were put forward:What were the dynamics of emotional, external, and intrapsychic stress before the pandemic and during different waves among athletes in Poland, Romania, and Slovakia (country split and wave split)?What were the dynamics of perceived emotional tension, external stress, and intrapsychic stress in the total sample of athletes (regardless of country) throughout the research periods (until the fourth wave of the pandemic)?What are the differences in the frequency of using strategies of coping with stress among athletes in the first and fourth waves of the pandemic?

## 2. Materials and Methods

### 2.1. Design

Data collection was carried out from November 2019 to January 2020 (the pre-pandemic period) as well as during or in the proximity of the first, second, third, and fourth waves of the COVID-19 pandemic (see Table 1). For example, in Romania, the peak of illnesses was recorded on 18 November 2020 (for the second wave) and on 25 March 2021 for the third wave. It is worth mentioning, also, that the first case of the SARS-CoV-2 coronavirus appeared in Poland on 4 March 2020, in Romania on 25 February, and in Slovakia on 6 March 2020. When completing the Perception of Stress Questionnaire, the instruction was as follows: “Please describe your thoughts, behaviors, fears and hopes as you have experienced them lately (in the last few weeks) and currently”. Data collection was carried out in Poland, Romania, and Slovakia and was concluded at the beginning of 2022. It is important to emphasize that in November 2019 (when data collection began) nobody knew about COVID-19. The original research idea of analyzing the stress experienced by athletes from Poland, Romania, and Slovakia at time t_0_ (the moment they completed the survey) and the coping strategies used were restructured to conduct a longitudinal study, in order to observe the dynamics of athletes’ perceived stress (during different waves of the pandemic) and coping strategies used to deal with stress.

### 2.2. Participants

A total of 2020 professional athletes took part in the study, practicing various sports disciplines: handball, soccer, martial arts (kickboxing, judo, fencing, karate, MMA, taekwondo), rugby, basketball, athletics, aerobic and artistic gymnastics, volleyball, tennis, and swimming (a total of sixteen sports disciplines in each country). The inclusion criteria were a career of at least two years of training in a specific sport branch, under the supervision of a coach and a minimum age of 18 years (seniors). Athletes have been practicing the sports disciplines (in the entire sample) for an average of 8.3 years. About 82% of the participants achieved local/regional level performances, approximately 12% registered national performances (being national champions, vice-champions, or being part of the national teams in the branch of sport practiced), while about 6% obtained international results (at World or European level, only martial arts athletes). In each research period/wave of the pandemic and in each country, athletes having local/regional, national, and international performances were investigated. No missing values were identified due to the online survey/submission in which all items had to be rated. In the preliminary analysis of the data, using stem and leaf, eighteen cases (in the total sample) were recognized as outliers and excluded from further investigation. Thus, we retained 2020 athletes (from the total sample of 2038 eligible athletes). It is relevant, also, to highlight that approximately 70% of the athletes tested in each research period/in every wave of the pandemic (in each country) were tested also in the pre-pandemic period. Table 1 shows the athletes’ descriptive statistics divided by gender, age, and data collection period.

The data collection carried out in the third wave of the pandemic (from April to the end of June 2021) presented a smaller number of people surveyed. To avoid a sample size reduction, 3rd wave data were not included in subsequent statistical analyses.

### 2.3. Instruments

Personal data were collected using an ad hoc questionnaire regarding personal and sociodemographic data. It comprised four items measuring the participants’ age, gender, years of training, and sport type.

Stress was measured using the *Perception of Stress Questionnaire*, comprising 21 items which form three scales: *Emotional tension* (7 items, e.g., “I get nervous more often than I used to, and for no obvious reason”), *External stress* (7 items, e.g., “I feel drained by constantly having to prove I am right”), and *Intrapsychic stress* (7 items, e.g., “Thinking about my problems makes it hard for me to fall asleep”) [30]. The generalized stress level (total score) is the sum of the following scales: *Emotional tension*, *External stress*, and *Intrapsychic stress*. Participants answer each item on a five-point Likert-type scale from 1 (definitely disagree) to 5 (definitely agree). The Cronbach’s α reliability coefficient in the Polish sample was as follows: emotional tension: from 0.75 to 0.81; external stress: from 0.68 to 0.74; intrapsychic stress: from 0.77 to 0.80. The Cronbach’s α reliability coefficient in the Romanian sample was as follows: emotional tension: from 0.59 to 0.79; external stress: from 0.65 to 0.82; intrapsychic stress: from 0.72 to 0.85. The Cronbach’s α reliability coefficient in the Slovak sample was as follows: emotional tension: from 0.68 to 0.78; external stress: from 0.63 to 0.75; intrapsychic stress: from 0.72 to 0.80. The Perception of Stress Questionnaire has been used in studies of athletes [30] including Romanian and Slovak athletes. The translation of the questionnaire in Romanian and Slovak language was carried out with the consent of the author (Makarowski Ryszard) and respecting the author’s recommendations. First, the original Polish version was translated into English and then translated into Polish by translators with psychological experience. The final Romanian and Slovak versions of the English version were created through retroversion, compared and used in the study (this procedure has been used in previous research [31]).

Using the Brief COPE questionnaire, we measured the strategies of coping with stress. It comprises 28 items covering 14 coping strategies: *self-distraction, active coping, denial, substance use, use of emotional support, use of instrumental support, behavioral disengagement, venting, positive reframing, planning, humor, acceptance, religion*, and *self-blame* (two items for each strategy) [32]. The participants indicate their frequency of using each coping strategy on a four-point Likert-type scale, from 1 (I have not been doing this at all) to 4 (I have been doing this a lot). In all data collection periods, the Cronbach’s α reliability coefficients for each subscale in the Polish, Romanian, and Slovak versions ranged from 0.48 to 0.94.

The 14 coping strategies of the Brief COPE questionnaire can be grouped in several ways. In the current study, we decided to divide them into three groups: emotion-focused strategies (emotional support, positive reframing, acceptance, religion, humor), problem-focused strategies, (active coping, planning, use of informational support), and dysfunctional strategies (venting, denial, substance use, behavioral disengagement, self-distraction, self-blame), according to the model by Su et al. [33]. This model was also used in other studies on athletes and other samples in many countries [34,35,36,37,38].

### 2.4. Procedure

Participants were informed about the study aim and procedure. They were also informed about the anonymity of the collected data and the right to withdraw their participation at any time without having to provide a reason. Informed consent was obtained from all participants. Furthermore, this study was conducted in accordance with the recommendations of the Declaration of Helsinki, the Polish Psychological Association’s Psychologist’s Code of Ethics, the Slovak Psychological Association, and the Romanian Psychological Association and it was approved by the Ethics Committee of the National University of Physical Education and Sports in Bucharest, Romania (ID: 1185).

### 2.5. Data Analysis

All the standard statistical analyses were conducted using the Statistica v. 13 software. Data were presented by means and standard deviations. Analysis of variance was carried out using Tukey’s T test for unequal sample sizes. Independent *t* test was also used. The statistical significance was set at a *p*-value of ≤0.05 and effect size (Cohen’s d) was interpreted as follows: ≤0.2, trivial; >0.2, small; >0.6, moderate; >1.2, large; >2.0, very large; >4.0, nearly perfect [39]. Considering η2 the range intervals were: 0.01, small effect; 0.06, medium; 0.14, large effect [40]. All variables were normally distributed, with skewness coefficients in absolute value being less than 1 [41]. The assumption of the equality of variance was verified by Levene’s test (*p* > 0.05, see Table 2).

## 3. Results

Table 2 shows the analysis of variance results with the pandemic stage, that is, the pre-pandemic period and all four subsequent pandemic waves, as the grouping variable. The analyses were carried out separately for each national subsample (country split). Considering the significant differences observed between the research periods/waves of the pandemic (in each country and for each subscale of the Polish, Romanian, and Slovak versions), d value ranged from 0.22 to 0.80 (the smallest effect sizes were observed in each country for the total score).

The obtained results show that the highest overall stress levels among Polish and Slovak athletes were reported during the fourth wave of the pandemic. Romanian athletes reported the highest overall stress in the pre-pandemic period. In all three countries, the highest intrapsychic stress levels were reported during the fourth wave and the highest external stress levels were reported before the pandemic. Emotional tension was the highest among Polish and Slovak athletes in the fourth wave and, among Romanian athletes, before the pandemic. η2 (the overall effect size) indicates, generally, small or moderate to small differences between the examined waves of the pandemic (considering athletes’ perceived stress). Only for intrapsychic stress were moderate to strong differences found (in Romanian and Slovak athletes).

The obtained data show that all stress dimensions significantly decreased during the first and second wave of the pandemic but increased significantly during the fourth wave (except external stress where no significant differences were found compared to the first two waves). The highest increase was observed for *intrapsychic stress.* Eta^2^ values (the overall effect size) are: 0.02 (for emotional tension), 0.03 (for external stress), respectively, 0.06 (for intrapsychic stress), emphasizing moderate to small (respectively medium) differences between the research periods/waves of the pandemic, taking into account (in each investigated wave) the total sample of athletes (regardless of country).

Table 3 shows the differences in perceived stress (between countries, in each wave of the pandemic wave split) among the surveyed athletes.

The highest level of stress in individual waves of the pandemic occurred in Slovakia. The lowest level of general stress was recorded in athletes from Romania (except for the tests performed before the pandemic). The overall effect size (η2) shows, generally, small or moderate to small differences between the three countries when talking about athletes’ perceived stress, in each examined wave of the pandemic.

To examine whether and how the frequency of using coping strategies by athletes changed between the first and fourth waves of the pandemic, independent t-test was carried out. The results are shown in Table 4.

Analyzing the dynamics of coping strategy use between the first and fourth waves of the COVID-19 pandemic, it can be observed that *emotion-focused strategies* became less frequent among Polish athletes. Regarding individual coping strategies, *behavioral disengagement* and *venting* became more frequent, while *planning*, *positive reframing*, and *humor* became less frequent. No significant changes in the frequency of the other individual coping strategies were observed.

Among Romanian athletes, *dysfunctional strategies* became more frequent. Regarding individual coping strategies, a decrease in the frequency of using *active coping* and an increase in using *behavioral disengagement* were observed. Neither of the other individual coping strategies was used significantly more frequently in the fourth wave.In the Slovak athlete subsample, the frequency of using *problem-focused strategies* and *dysfunctional strategies* increased. Regarding individual coping strategies, *active coping, planning*, *acceptance*, and *venting* became more frequent. Neither of the other individual coping strategies was used significantly less or more frequently in the fourth wave.It is worth noting that *dysfunctional strategies* became noticeably more frequent in each national subsample during the fourth wave (this difference was not statistically significant in the Polish athlete subsample).

## 4. Discussion

The emergence of the COVID-19 pandemic changed the structure and functioning of the world as we know it, permanently and suddenly. Such a significant threat has not been experienced by many European countries for a long time. The coronavirus disease has impacted nearly all aspects of human functioning. Numerous strains have increased the intensity of experienced stress and initiated the activization of coping strategies [42,43,44]. Social contacts, working, transport, spending free time, and engaging in physical activity have changed noticeably. For professional athletes, limiting the opportunities for training and canceling or delaying sports events represented a significant challenge [45]. These situations occurred due to the lockdown periods (as preventive measures for reducing COVID-19 spread), because of the infections (or the fear of infection) of athletes, coaches, sports managers, and organizers of sports competitions. In an attempt to identify most of the infected athletes worldwide before August 2020 (according to gender, age, symptoms, sport level, or location of the contraction of infection), researchers found 521 COVID-19-positive athletes [46]. It seems that most infected athletes practiced soccer and basketball (as authors asserted, the cases do not represent all of the infected athletes). Considering that globally, as of 5:44 p.m. CEST, 24 August 2022, there have been 595,219,966 confirmed cases of COVID-19 reported to the World Health Organization (WHO) [47], but it is very difficult to identify the number of COVID-19 infections among athletes (more so as there are professional athletes, amateur, college, junior, or senior athletes and some of them were asymptomatic).

Significantly restricted training opportunities, canceled or delayed events, and reduced income have compounded the universal concerns about one’s own health and the health of one’s family. Thus, the aim of the current study was to identify the stress level dynamics during the first four waves of the COVID-19 pandemic, in Polish, Romanian, and Slovak athletes, and to establish the changes in coping strategies between the first and fourth waves. The dependent variable was the level of stress under study, namely emotional tension, external stress, and intrapsychic stress. The independent variables (the variable playing the role of IVs) were three countries: Poland, Slovakia, and Romania; the research period: before the pandemic and the first, second, and fourth wave of the pandemic. An additional dependent variable was the way of coping with stress. The results revealed that in all three countries, the highest intrapsychic stress levels were reported during the fourth wave and the highest external stress levels were reported before the pandemic. Further, the coping strategies used by the athletes in the fourth wave were more dysfunctional than during the first wave.

Perceived stress levels among athletes differed depending on the country. The highest level of stress in individual waves of the pandemic was reported by Slovak athletes, while the lowest level of general stress was registered in athletes from Romania (except for the tests performed before the pandemic). Small or moderate to small differences were observed between the three investigated countries, when talking about athletes’ experienced stress (in individual waves). The significant differences found (between countries) could be related to numerous variables: the intensity of the pandemic, the current economic and political situation in the country, and employment stability. The *Human Development Report*, which indicates the quality of life in a given country [48], is also important in this context.

In all countries, there was a noticeable trend in the overall pre-pandemic stress levels decreasing and remaining at a lower level throughout the first and second wave of the pandemic, before increasing during the fourth wave. Significant differences were observed in each investigated country (studied separately), between the waves of the pandemic, as well as small or moderate to small effect sizes (for emotional tension and external stress). In the case of intrapsychic stress, a moderate to strong effect size (Eta^2^) was found in Romanian and Slovak athletes. Furthermore, when the three countries were studied together (the total sample), moderate to small (respectively, medium for intrapsychic stress) differences between the research periods/waves of the pandemic were highlighted. Similar results were observed for martial arts practitioners from Poland and Romania and also in athletes practicing various sports disciplines (non-martial-arts athletes) from Poland, Slovakia, and Romania, with stress levels decreasing during the height of the pandemic (during the lockdown and first wave), compared to the pre-pandemic period [30]. However, there are also studies that underline that about one month after the beginning of the lockdown (first wave), perceived stress increased in Italian athletes from individual and team sports [13] (martial arts athletes were not included in the sample).

Such differences (a significantly lower level of stress during the first three waves of the pandemic) can be explained when considering the psychological and social mechanisms behind the observed trend—it can be assumed that several phenomena co-occurred. First, habituation led to a lower intensity of reaction to a repeated stimulus. Due to a long-term presence of a constant stimulus, the stress reaction becomes reduced and, in time, it may become extinct [49,50,51]. Habituation has an adaptive function, as it allows for economical use of the individual’s emotional and cognitive resources. Stress could have also decreased due to the cancellations or delays of upcoming sports events [52]. For many athletes, participation in competition involves intense psychological stress. In various sports disciplines (e.g., volleyball, tennis, track and field, cycling, boxing, soccer), higher perceived stress levels were observed upon resumption of competitions [14]. Further, for example, in swimmers, a significant release of stress hormones was observed, as a result of physical and mental stress associated with sports competition [53]. Cancelling or delaying such events may reduce psychological tension. Another reason for lowered stress levels could be the reduction in intensity of training and work, with the resulting rest and isolation allowing for a regeneration of psychological resources [54,55]. Due to training and competitions being limited, the perceived level of rivalry also decreased, as all athletes found themselves in similar circumstances [56,57]. It is worth underlining that the restrictions and possibilities in each time span (and in each investigated country) were relatively the same—during the lockdown period, coaches and athletes worked exclusively on different remote learning platforms (online) at home. When the conditions relaxed, athletes were able to practice outdoors, on sports fields or in parks, respecting the measures of social distancing. There are differences between sports branches. Some players, such as runners, were able to follow more easily the training plan during the COVID-19 pandemic (there was, almost, no break in the training). Sports competitions were also organized in Poland, Romania, and Slovakia (and televised) but without spectators (for months). Only coaches and athletes had access to the competition hall/area and they were previously tested against COVID-19. It is also important to mention that the vaccination campaign began (in the three countries) at the end of 2020—in Romania, on 27 December 2020. For example, during the third wave of the pandemic (on 7 April 2021), according to the National Institute of Public Health [58], only 1.288.487 Romanians (about 6.5% from the population) were vaccinated with both doses.

Sport is a stress-generating environment, as unpleasant remarks coming from supporters and noise from the stands (in many sports) increase athletes’ stress levels [59,60]. The absence of spectators (during the first waves of the pandemic) can, therefore, reduce experienced stress in athletes. Here, also, we emphasize differences between sports disciplines, with team sports athletes feeling less negative stress [61] and reporting less anxiety and depression than in individual sports [62], while novice performers registered higher perceived stress than top athletes (potentially reflecting their less-adapted coping resources) [13]. Further, athletes with high athletic identity are less prone to higher levels of psychological distress compared to athletes with low athletic identity [61].

The pandemic has brought into focus the fundamental human issues of health and survival. Among stressors for athletes were the fear of COVID-19 infection (the fear of health deterioration), weight change, exercising at home, monthly income perception, and damaged performance in COVID-19 infection [63] (it is important to mention that athletes’ trait anxiety values were below average). The risk of infection when rigorously following the hygiene and social isolation protocols is minimal and other life events are unable to cause intense stress reactions in the context of a global pandemic. When fundamental issues, such as health and survival, are threatened, people appraise everyday problems differently [64,65]. Moreover, the reduction in sports events resulted in a significant reduction in athletes’ media exposure, which could have been a major source of pressure [66,67]. The increase in stress during the fourth wave of the pandemic could have been caused by the depletion of personal resources and poorer adaptation to the permanent conditions of the pandemic [68,69], as well as chronic fatigue syndrome [70]. All limitations and restrictions accumulated over time may cause higher levels of stress and greater health concerns. Further, athletes’ worsening economic conditions and uncertain prospects for the future may be significant [71].

Emotional and external stress among athletes was lower during the first, second, and fourth wave than during the pre-pandemic period. However, a very high increase in intrapsychic stress was observed during the fourth wave. Increased intrapsychic stress is a consequence of prolonged negative events, with which the individual has not coped effectively (i.e., has used dysfunctional coping strategies) [30]. Past, present, as well as future anticipated events may be sources of stress [72]. In the context of a prolonged pandemic, this results in an accumulation of perceived stress.

The smaller number of people surveyed in the third wave of the pandemic resulted in the exclusion of this group of people due to the possibility of distorting the results of the research.

The long duration of the pandemic has also impacted strategies of coping with stress. On the one hand, athletes have learned which coping strategies are effective. On the other hand, permanent functioning during the pandemic may have modified the employed strategies. To identify changes in coping strategies among athletes, we compared them between the first and the fourth waves of the pandemic. In the Polish athlete subsample, we observed a decrease in the frequency of using *emotion-focused strategies*. It is assumed that emotion-focused strategies are ineffective in the long term, though they may negate the consequences of stress in some situations [73,74]. In the Romanian athlete subsample, the use of *dysfunctional strategies* increased in frequency. In the long term, using these coping strategies can be associated with a risk of depression, anxiety, and eating disorders [75,76]. Among Slovak athletes, the frequency of using *problem-focused strategies* and *dysfunctional strategies* increased simultaneously. Using *problem-focused strategies* allows for coping with difficult situations in the most effective way [77].

Our study revealed a significant trend in coping strategy use among athletes. Comparing the frequency of using coping strategies, we observed an increase in using *dysfunctional strategies* in each country. Aggregated results for individual coping strategies show that, in each country, the frequency of using *dysfunctional strategies* during the fourth wave of the pandemic was higher than in the first wave. Long-term functioning in stressful situations may reduce personal resources. Thus, seeking easier solutions for a difficult situation, athletes more frequently used *dysfunctional strategies.* Long-term use of such strategies carried with it a risk of depression and worsened health [78]. However, these strategies can be modified with appropriate psychological intervention [79]. Considering elite athletes, as well as physical education students practicing sports most often, researchers highlight the important role of cognitive and behavioral strategies in coping with the stress generated by the COVID-19 pandemic [80]. It was found that “the sports level depended on the strategies of coping with the stress of the COVID-19 pandemic more strongly than gender”.

Regarding individual coping strategies, in the Polish athlete subsample, the frequency of using *behavioral disengagement* and *venting* increased, while the frequency of using *planning*, *positive reframing*, and *humor* decreased. The first two of these are *dysfunctional strategies.* Increasing the frequency of using these strategies decreases the probability of effectively coping with stress. Even if in the short term, they may prove useful in reducing perceived stress, we cannot promote them, given their known long-term effects [30]. *Positive reframing* and *humor* are *emotion-focused strategies.* They are not effective in the long term for the subsequent pandemic waves. Decreasing the frequency of using these strategies decreases the probability of effectively coping with stress and minimizing its effect on wellbeing. The long duration of the pandemic was related to a decrease in the frequency of using *planning*, a *problem-focused strategy.* The unpredictability of the pandemic, together with a lack of control over many aspects of life, may have caused a decrease in using this strategy in the perspective of the pandemic’s increasing duration.

In the Romanian athlete subsample, we noticed an increase in the frequency of using the individual strategy of *behavioral disengagement*, which is a *dysfunctional strategy*. It can be assumed that, similar to Polish athletes, the lack of control over the situation could have caused an increase in the frequency of using this strategy. However, a different pattern was observed among the Slovak athletes. They reported an increase in using *active coping*, *planning* (*problem-focused strategies*), and acceptance (emotion-focused strategy), as well as a decrease in venting (dysfunctional strategy).

Permanent use of *dysfunctional strategies* is ineffective and is related to a risk for depression [81]. The noticeable increase in the frequency of using these coping strategies by athletes during the fourth wave of the pandemic, together with the increase in intrapsychic stress, should alert coaches and sport psychologists to the ways in which professional athletes modify their use of available coping strategies. Close cooperation with a sports psychologist and coach is essential in order to promote the most effective coping strategies for a given person [82]. Along with medical practitioners, members of the multidisciplinary team should work towards minimizing the strain experienced by athletes [83]. In order to reduce athletes’ distress, specialists could use the so-called *internal* techniques (breathing and meditation, self-control techniques) [84], inner monologue (positive self-talk) increasing self-confidence, analytical relaxation, and autogenic training; self-monitoring of emotional reactions [85] could teach athletes positive conflict resolution strategies and guide them to get involved in motor and mental activities, which gives them great satisfaction [86]. Not least, specialists can use written emotional disclosure (WED) to support athletes during the COVID-19 pandemic and to promote their mental health [87] and the 4Ds for dealing with distress, an ultra-brief single session, which unifies strategies and exercises for problem-solving, emotion regulation, and for increasing resilience (restoring wellbeing) [88].

The present study has some limitations. The authors relied on self-report measures, supposing a possible recall bias and/or the issue of possible desirable answers (when talking about explicit evaluations), aspects being known [89] (however, the large number of athletes tested represents a strength of the study). Further, the results may be different if junior athletes were investigated, athletes from other countries, practicing a single sport discipline, if athletes were examined separately, according to the level of training, as well as according to their property status (these are relevant questions for future research). Moreover, the Cronbach’s α reliability coefficients presented a range of low to very high level of reliability for the strategies of coping with stress, which should also be considered when interpreting the results of this study. Finally, even if there is a reciprocal relationship between stress and anxiety (the two dimensions having a mutual influence on each other), other investigation tools are recommended for anxiety, capturing the link between the anxiety of athletes (state anxiety and/or trait anxiety) and the size of a pandemic in a given country. This can be the subject of further research.

## 5. Conclusions

The conclusions of the current study, carried out in three countries, showed that the direct consequences of the pandemic are not related to an increase in perceived stress among athletes. Overall, stress levels during the fourth wave of the pandemic were not higher, in all countries, than during the pre-pandemic period. However, an increase in intrapsychic stress was noticeable between the first two waves and the fourth wave of the pandemic. The research is underlining the importance of athletes’ experienced stress (which can influence, also, their anxiety level), capturing the dynamics of perceived emotional tension, external stress, and intrapsychic stress in athletes, before and throughout different waves of the COVID-19 pandemic.

Using constructive coping strategies allows for reducing the perceived stress. Using these strategies led to lower stress levels. Coaches and sport psychologists should continuously monitor stress levels among athletes, together with their coping efforts, in order to promote effective coping strategies. As the pandemic may have long-term consequences, it is particularly important to monitor athletes’ psychological wellbeing also after its end, in a post-COVID-19 world.

## Figures and Tables

**Table 1 healthcare-10-01770-t001:** Descriptive statistics and data collection timetable.

Country	Research Period	*n*	Men	Women	M_age_	SD
Poland	Before the Pandemic 11.2019–1.2020	314	186	128	22.85	3.25
Romania		221	133	88	21.86	3.60
Slovakia		91	51	40	23.27	2.28
Poland	1st wave 4–6.2020	134	84	50	26.40	4.98
Romania		145	85	60	25.22	5.65
Slovakia		111	67	44	23.73	6.05
Poland	2nd wave 10–12.2020	63	31	32	24.25	4.77
Romania		171	112	59	20.82	5.77
Slovakia		99	48	51	23.32	9.10
Poland	3rd wave 4–6.2021	76	40	36	23.39	3.69
Romania		99	57	42	21.34	3.84
Slovakia		94	40	54	22.60	5.28
Poland	4th wave 10–12.2021	127	60	67	26.16	4.49
Romania		174	103	71	22.21	5.84
Slovakia		101	49	52	23.94	3.91

M, mean age in years; SD, standard deviation.

**Table 2 healthcare-10-01770-t002:** Stress levels in Polish, Romanian, and Slovak athletes before the COVID-19 pandemic and during the first, second, and fourth waves.

**Country**	Pandemic	*n*	Emotional Tension		External Stress		Intrapsychic Stress		Total Score	
			M	SD	M	SD	M	SD	M	SD
**Poland**	0	314	17.72	5.66	18.52	5.40	14.20	5.37	50.46	16.66
	1st wave	134	17.05	5.50	17.08	5.07	12.32	4.72	46.48	16.70
	2nd wave	63	15.73	6.30	17.49	5.99	13.31	5.33	46.54	16.61
	4th wave	127	18.19	6.65	17.60	5.58	15.83	5.67	51.64	16.31
	F		2.87 (*p* = 0.035)		2.61 (*p* = 0.051)		9.98 (*p* = 0.001)		3.97 (*p* = 0.010)	
	Levene’s test		2.98 (*p* = 0.531)		1.34 (*p* = 0.254)		1.47 (*p* = 0.211)		73.63 (*p* = 0.237)	
	differences		0:2 *; 1:2 *; 4:(1.2) *		0:1 *		0:1 **; 0:4 *; 4:(1.2) **		0:(1.2) ***; 4:(1.2) ***; 0:4 *	
	Eta^2^ (η2)		0.01		0.01		0.04		0.015	
**Romania**	0	221	18.41	5.89	19.68	5.53	15.50	4.82	53.60	14.69
	1st wave	145	15.92	6.12	16.50	5.42	13.48	4.97	45.91	15.23
	2nd wave	171	15.39	5.61	16.82	5.37	13.10	4.92	45.32	15.25
	4th wave	174	17.17	7.24	16.50	5.53	16.28	5.89	49.95	16.86
	F		8.79 (*p* < 0.001)		16.28 (*p* < 0.001)		14.95 (*p* < 0.001)		11.83 (*p* = 0.002)	
	Levene’s test		7.68 (*p* = 0.082)		6.03 (*p* = 0.092)		4.15 (*p* = 0.068)		2.88 (*p* = 0.544)	
	differences		0:(1.2) **; 0:4 *; 4:(1.2) *		0:(1.2.4) ***		0:(1.2) **; 4:(1.2) **		0:(1.2.4) ***; 1:4 **; 2:4 **	
	Eta^2^ (η2)		0.04		0.07		0.07		0.066	
**Slovakia**	0	91	17.97	4.82	20.25	4.46	15.75	4.12	53.98	13.06
	1st wave	111	17.44	5.00	18.90	4.11	15.09	4.15	51.43	12.17
	2nd wave	99	16.42	6.00	18.47	5.65	14.45	4.91	49.35	16.14
	4th wave	101	18.74	6.21	18.68	5.11	18.48	5.12	55.91	14.87
	F		3.11 (*p* = 0.026)		2.55 (*p* = 0.054)		14.99 (*p* < 0.001)		4.69 (*p* = 0.003)	
	Levene’s test		6.41 (*p* = 0.513)		4.75 (*p* = 0.083)		2.85 (*p* = 0.080)		5.93 (*p* = 0.062)	
	differences		0:2 *; 2:4 **; 1:4 *		0:(1.2.4) *		0:(2.4) *; 4:(1.2) **		4:(1.2) ***	
	Eta^2^ (η2)		0.02		0.02		0.11		0.031	

0—before the pandemic. * *p* ≤ 0.05. ** *p* ≤ 0.01. *** *p* ≤ 0.001. η2 = 0.06 indicates a medium effect.

**Table 3 healthcare-10-01770-t003:** Stress levels in Polish, Romanian and Slovak athletes before the COVID-19 pandemic and during the first, second, and fourth waves (analysis of differences between countries).

Pandemic	Country	N	Emotional Tension		External Stress		Intrapsychic Stress		Total Stress	
			M	SD	M	SD	M	SD	M	SD
Before pandemic	Poland	314	17.72	5.66	18.52	5.41	14.20	5.37	50.46	16.66
	Romania	221	18.41	5.89	19.68	5.53	15.50	4.82	53.60	14.69
	Slovakia	91	17.79	4.82	20.25	4.46	15.75	4.12	53.98	13.06
	F		0.88		5.19		6.00		3.97	
	Levene’s test		3.60 (*p* = 0.058)		4.78 (*p* = 0.088)		7.70 (*p* = 0.051)		7.57 (*p* = 0.101)	
	differences		NS		1:3 *		1:2 *		1:2 *; 1:3 *	
	Eta^2^ (η2)		0.003		0.02		0.02		0.01	
1st wave	Poland	134	17.05	5.50	17.08	5.07	12.32	4.72	46.48	16.70
	Romania	145	15.92	6.12	16.50	5.42	13.48	4.97	45.91	15.23
	Slovakia	111	17.44	5.00	18.90	4.11	15.09	4.15	51.43	12.17
	F		2.61		7.71		10.64		6.99	
	Levene’s test		10.39 (*p* = 0.052)		5.98 (*p* = 0.070)		2.62 (*p* = 0.59)		6.93 (*p* = 0.065)	
	differences		1:2 *; 2:3 *		1:3 *; 2:3 ***		1:2 *; 2:3 *		1:3 ***; 2:3 ***	
	Eta^2^ (η2)		0.01		0.04		0.09		0.03	
2nd wave	Poland	63	15.73	6.30	17.49	5.99	13.31	5.33	46.54	16.61
	Romania	171	15.39	5.61	16.82	5.37	13.10	4.92	45.32	15.25
	Slovakia	99	16.42	6.00	18.47	5.65	14.45	4.91	49.35	16.14
	F		0.988		2.73		2.40		10.38	
	Levene’s test		0.98 (*p* = 0.37)		2.76 (*p* = 0.65)		2.36 (*p* = 0.095)		2.25 (*p* = 0.101)	
	differences		2:3 *		2:3 *		1:3 *; 2:3 *		1:3 *; 2:3 **	
	Eta^2^ (η2)		0.005		0.01		0.01		0.01	
4th wave	Poland	127	18.19	6.65	17.60	5.58	15.83	5.67	51.64	16.31
	Romania	174	17.17	7.24	16.50	5.53	16.28	5.89	49.95	16.86
	Slovakia	101	18.74	6.21	18.68	5.11	18.48	5.12	55.91	14.87
	F		1.90		5.27		7.01		4.36	
	Levene’s test		2.54 (*p* = 0.079)		1.22 (*p* = 0.295)		1.85 (*p* = 0.158)		2.27 (*p* = 0.103)	
	differences		2:3 *		2:3 *		1:3 *; 2:3 *		1:2 *; 1:3 **; 2:3 ***	
	Eta^2^ (η2)		0.01		0.02		0.03		0.02	

NS, not significant. 1-Poland, 2-Romania, 3-Slovakia. * *p* ≤ 0.05. ** *p* ≤ 0.01. *** *p* ≤ 0.001. η2 = 0.06 indicates a medium effect.

**Table 4 healthcare-10-01770-t004:** Strategies of coping with stress among Polish, Romanian, and Slovak athletes during the first and fourth waves of the pandemic.

Country	Coping Strategies	1 Wave		4 Wave		t	p	d
		M	SD	M	SD			
**Poland**	Active coping	4.48	1.32	4.27	1.16	1.391	0.165	0.17
	Planning	4.61	1.35	4.29	1.26	1.992	0.047	0.24
	Positive reframing	3.28	1.44	2.67	1.44	3.447	0.001	0.42
	Acceptance	4.14	1.37	4.11	1.24	0.177	0.860	0.02
	Humor	2.82	1.55	2.45	1.35	2.042	0.042	0.25
	Religion	2.38	2.12	1.68	1.87	2.835	0.005	0.35
	Emotional support	3.33	1.83	3.54	1.67	–0.988	0.324	0.12
	Use of informational support	3.20	1.81	3.17	1.43	0.154	0.878	0.02
	Self-distraction	3.16	1.45	3.29	1.46	–0.726	0.468	0.09
	Denial	1.60	1.46	1.70	1.37	–0.540	0.590	0.07
	Venting	2.51	1.39	2.84	1.23	–2.029	0.043	0.25
	Substance use	0.98	1.57	0.77	1.23	1.225	0.222	0.15
	Behavioral disengagement	1.00	1.13	1.43	1.18	–3.027	0.003	0.37
	Self-blame	2.45	1.66	2.65	1.64	–0.983	0.326	0.12
	Emotion-Focused strategies	15.94	5.30	14.45	4.82	2.390	0.018	0.29
	Problem-Focused strategies	12.29	3.68	11.73	2.62	1.428	0.154	0.18
	Dysfunctional strategies	11.70	4.94	12.66	4.78	–1.622	0.106	0.20
**Romania**	Active coping	4.85	1.19	4.55	1.39	2.022	0.044	0.23
	Planning	4.53	1.16	4.41	1.41	0.825	0.410	0.09
	Positive reframing	3.31	1.52	3.14	1.55	0.980	0.328	0.11
	Acceptance	4.07	1.35	4.05	1.51	0.141	0.888	0.02
	Humor	3.23	2.00	3.06	2.07	0.733	0.464	0.08
	Religion	2.63	1.98	2.39	1.87	1.113	0.267	0.12
	Emotional support	3.90	1.63	3.84	1.75	0.315	0.753	0.04
	Use of informational support	3.63	1.63	3.83	1.73	–1.093	0.275	0.12
	Self-distraction	3.24	1.48	3.47	1.70	–1.313	0.190	0.15
	Denial	1.41	1.65	1.75	1.72	–1.813	0.071	0.20
	Venting	2.96	1.81	3.29	1.85	–1.608	0.109	0.18
	Substance use	0.55	1.27	0.69	1.42	–0.902	0.368	0.10
	Behavioral disengagement	0.94	1.34	1.27	1.61	–1.991	0.047	0.22
	Self-blame	2.65	1.82	2.91	1.81	–1.280	0.202	0.14
	Emotion-Focused strategies	17.15	5.13	16.49	5.55	1.094	0.275	0.12
	Problem-Focused strategies	13.01	2.83	12.80	3.53	0.578	0.564	0.07
	Dysfunctional strategies	11.75	5.89	13.39	6.63	–2.318	0.021	0.26
**Slovakia**	Active coping	4.03	1.37	4.51	1.32	–2.388	0.018	0.36
	Planning	3.63	1.43	4.21	1.47	–2.666	0.008	0.40
	Positive reframing	3.04	1.41	2.87	1.40	0.852	0.395	0.13
	Acceptance	3.55	1.29	4.16	1.48	–2.849	0.005	0.44
	Humor	1.93	1.77	2.10	1.93	–0.611	0.542	0.09
	Religion	1.91	1.96	1.57	1.90	1.194	0.234	0.18
	Emotional support	3.82	1.46	3.81	1.56	0.033	0.974	0.00
	Use of informational support	3.24	1.44	3.70	1.66	–1.946	0.053	0.30
	Self-distraction	3.06	1.37	3.47	1.59	–1.802	0.073	0.28
	Denial	1.12	1.16	1.25	1.45	–0.658	0.511	0.10
	Venting	2.69	1.20	3.08	1.38	–2.003	0.047	0.31
	Substance use	0.70	1.28	0.69	1.33	0.076	0.940	0.01
	Behavioral disengagement	3.55	1.26	3.88	1.53	–1.520	0.130	0.23
	Self-blame	2.06	1.62	2.18	1.65	–0.486	0.628	0.07
	Emotion-Focused strategies	14.25	4.62	14.50	4.55	–0.360	0.719	0.05
	Problem-Focused strategies	10.90	2.90	12.42	3.07	–3.377	0.001	0.51
	Dysfunctional strategies	13.18	3.70	14.55	4.43	–2.184	0.030	0.34

M, mean. SD, standard deviation. d, Cohen’s effect size.

## Data Availability

Data are available upon request to the contact author.

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
