# Peer review of "Coping Strategies and Perceiving Stress among Athletes during Different Waves of the COVID-19 Pandemic—Data from Poland, Romania, and Slovakia"

_healthcare, 2022, doi:10.3390/healthcare10091770_

Round 1
Reviewer 1 Report
Dear authors,
thank you for this well done study. It would be interesting to know whath the original aim of your study was. In November 2019 we didn't know about COVID, when you began the data collection. Could this original aim play a role? In addition it would be interesting to analyze the influences of age, gender, sports type or duration of competition membership, showed by a regression analysis e.g. Further you only have a few typos that should be corrected. In table 1, you could add in the description "age in years" to understand the numbers at the first sight. In table 2, you provide highly significant differences as well as strong Eta² in most of the variables between all measurements. Why could these differences be? What were the restrictions and possibilities in each time span? Please revise the discussion by these points.
Best regards!
Author Response
Dear reviewer Dear professor
Thank you for your time and insightful comments on article. Suggestions for changes and questions allowed us to improve our article. The changes made to the text are marked in green.
Regards
Authors
Reviewer 2 Report
Thank you for giving me the opportunity to review this manuscript which described a survey-based study on Perceiving stress among athletes in the first, second, third, and 2 fourth wave of the COVID-19 pandemic. Generally, I find this research interesting and within the growing number of investigations regarding COVID-19 impacts on mental health. The methodical aspects are well designed, and the results section is correctly conceived with the recommendation for convenient reaction focused on the wellbeing of athletes.
I would recommend the authors to improve the text by addressing the following comments:
Abstract:
(1) Line 29, authors are recommended to introduce COVID-19 for the first time.
(2) Authors are recommended to add a brief description of statistical analysis methods used to analyze the data of the study within the abstract.
Introduction:
(1) Line 48, the authors are recommended to re-introduce COVID-19.
(2) Paragraph 2, the authors should elaborate more on how the COVID-19 pandemic impacted the general population mental health and sport activities.
(3) Lines 73-74, the authors stated that “Athletes’ material conditions naturally influence their functioning.” What is the material condition described here? I think the authors meant to say mental status? If the word “material” was a typo error, please correct it.
(4) Lines 77-78, the authors stated that “Effective coping strategies reduce stress experienced by athletes during the COVID- 19 pandemic [12,13].” What coping strategies were the authors referring to here? Please clarify and elaborate more on this.
(5) Lines 87-89, the authors are recommended to paraphrase the quoted statement “As authors mentioned - “perception of control can change over time, which is reflected in structural developmental changes…and the extent to which well-learned coping responses are effectively executed in relation to environmental demands” (p. 63).” and cite it properly.
(6) Throughout the reading of the introduction, I find that the introduction needs re-organization to improve logical flow of ideas. Paragraphs talking about stress should be ordered in logical manner to achieve coherence.
Methods:
(1) Design: specify the dates for first, second, third, and fourth waves of the COVID-19 pandemic within the text or refer to that within Table 1.
(2) Furthermore, initially how many athletes were invited to participate in the study? How many were excluded? Were there any missing data? How missing data were treated- what type of imputation method was used? What was the response rate? All these data/information were not described.
Results:
(1) I find the result section is well summarized and included only the salient features of the study rather than repeating what is already presented in the tables.
Discussion:
(1) Somewhere in paragraph 1, the authors are recommended to summarize the key findings from the research.
(2) Lines 273-274, the authors stated that “For professional athletes, limiting the opportunities for training and canceling or delaying sports events represented a significant challenge [46].” The authors are recommended to elaborate more on what are the causes for delaying or canceling sports activities? Is it the lockdown or infections of athletes/coaches or lack of preventive measures …etc. The authors are recommended to list any available data about the number COVID-19 infections among the athletes during the pandemic in the world or within the study countries.
(3) Lines 285-287, The authors stated that “In all countries, there was a noticeable trend of the overall pre-pandemic stress levels decreasing and remaining at a lower level throughout the first, second, and third wave of the pandemic, before increasing during the fourth wave.” The authors are recommended to support their statement with relevant reference(s).
(4) Since data after the pandemic was collected retrospectively, authors are recommended to acknowledge that recall bias is one of the study limitations.
Author Response

(The authors gave the same response as above.)

Reviewer 3 Report
Dear Authors,
The article is certainly interesting and relevant.
I have a few comments.
Introduction
Line 77-94
This does not apply to the topic of the work. It shouldn't be discussed that widely. May be removed or shortened to two sentences.
Line 95-101
I don't see a close relationship. I propose to delete.
Line 113-129
This should be in the discussion.
The research questions should include the causes of stress in different countries and the assessment of their differences.
What was the reason, the criterion for the inclusion of these three countries?
The article should indicate its purpose.
Materials and Methods
Information on the exact months of these stages should be added. Were they everywhere at the same time?
Stress is experienced differently depending on the discipline. In team sports it is smaller than in individual sports. This should be included in the work. Additionally, some players, such as runners, could follow the training plan. He was not addicted to disease. So there was no break in the training. This is an important point.
Why were there fewer respondents in the second and third waves? Shouldn't the test subjects be excluded (fourth wave)?
What was the training level of the athletes? Were they professionals? Are the juniors? Was their property status dependent on performance?
Information about the stress coping assessment should be added to the title.
Discussion
Line 280-283
Variables should be indicated. This is the main differentiating factor apart from the place of residence.
Line 295-299
Which players did this concern? Are they all?
Line 300-310
What exactly were the causes of stress in these athletes? Are these listed? Did they concern everyone? Is it subjective information of the authors?
Line 311-317
Were these changes statistically significant or just trends? It is not marked on the chart.
I can't find this in the results.
The talk is about changes in stress and coping. I miss a factor that influences it. Stress assessment and its type. Again, information about counseling should be indicated in the title of the work and in the introduction, because it is widely discussed in the results and discussions.
Figure 1 is interesting, but why, for example, differences between countries have not been identified? If the article talks about stress in different countries, the results should also be consistent with this division. The change of stressors in different countries and stages of the study would be interesting. Showing again what these stressors were and whether it was influenced by the political situation, the number of cases, deaths, and restrictions are important, and it seems to me that they are missing.
In my opinion, the correlation between e.g. the size of a pandemic in a given country and the anxiety of athletes should be investigated.
Information on why this stress varied across countries and during the stages of a pandemic is important and interesting.
What was the main cause of stress in these people? Infection, training break or something else?
The authors asked the question: ,, What were the dynamics of emotional, external, and intrapsychic stress before the pandemic and during its first, second, third, and fourth wave among athletes in Poland, Romania, and Slovakia? '' and showed the total results. This is a bit confusing as you can expect a country split as well.
Kind regards,
Reviewer

Author Response

(The authors gave the same response as above.)

Round 2
Reviewer 3 Report
Dear Authors,
This information should be added ex:
The first case of the Sars-Cov-2 coronavirus appeared in Poland on March 4, 2020, in Romania on February 25, and in Slovakia on March 6, 2021.
The essence of this information is to indicate the standardization of measurements.
I do not agree that people not participating in the third study should be included in the overall results. This distorts the overall statistic. Explaining that they did not have time to fill in the questionnaire during the lockdown is unlikely and certainly not suitable for research papers.
The results should be corrected.
Q (10) What was the training level of the athletes? Were they professionals? Are the juniors? Was their property status dependent on performance?
A: The changes were made as suggested by the reviewer.
These groups should be standardized and assessed differently. There is a difference between stress in competitive athletes and, for example, in juniors. They differ in many factors, such as property status, earnings or not in sports, etc.
, A: Changes were made in Table 3, and considering the interesting link between the anxiety of athletes and the size of a pandemic in a given country a different study can be made, investigating state anxiety and/or trait anxiety with different instruments. Although there is a reciprocal relationship between stress and anxiety (the two dimensions having a mutual influence on each other), we consider it better to use other investigation tools for anxiety. This can be the subject of further research.’’ - This information should be added to the limitations of the study.
Kind regards,
Reviewer
Author Response
Dear Reviewer Dear Professor
thank you for your comments. We have improved the article and the explanations are in a separate file.
Regards
authors